# Synergistic Effects of DHA and Sucrose on Body Weight Gain in PUFA-Deficient Elovl2 -/- Mice

**DOI:** 10.3390/nu11040852

**Published:** 2019-04-15

**Authors:** Anna M. Pauter, Alexander W. Fischer, Tore Bengtsson, Abolfazl Asadi, Emanuela Talamonti, Anders Jacobsson

**Affiliations:** 1Department of Molecular Biosciences, The Wenner-Gren Institute, Stockholm University, SE-10691 Stockholm, Sweden; annapauter0@gmail.com (A.M.P.); al.fischer@uke.de (A.W.F.); tore.bengtsson@su.se (T.B.); abbe.asaudi@su.se (A.A.); anders.jacobsson@su.se (A.J.); 2Department of Biochemistry and Molecular Cell Biology, University Medical Center; Hamburg-Eppendorf, 20246 Hamburg, Germany

**Keywords:** DHA deficiency, ELOVL2, DHA supplementation, high sucrose diet

## Abstract

The omega-3 polyunsaturated fatty acid docosahexaenoic acid (DHA) is implicated in the regulation of both lipid and carbohydrate metabolism. Thus, we questioned whether dietary DHA and low or high content of sucrose impact on metabolism in mice deficient for elongation of very long-chain fatty acids 2 (ELOVL2), an enzyme involved in the endogenous DHA synthesis. We found that Elovl2 -/- mice fed a high-sucrose DHA-enriched diet followed by the high sucrose, high fat challenge significantly increased body weight. This diet affected the triglyceride rich lipoprotein fraction of plasma lipoproteins and changed the expression of several genes involved in lipid metabolism in a white adipose tissue. Our findings suggest that lipogenesis in mammals is synergistically influenced by DHA dietary and sucrose content.

## 1. Introduction

Many studies implicate a role of the omega-3 polyunsaturated fatty acid (PUFA) docosahexaenoic acid (DHA, 22:6(n-3)) in the regulation of various metabolic processes such as β-oxidation, adipogenesis [1], lipogenesis [2], and glucose metabolism [3]. Therefore, dietary supplementation with DHA has been intensively studied in rodents and humans and has been proposed as a potential beneficial factor for the treatment of metabolic syndrome related diseases [4]. In addition, it has been postulated that depletion of DHA in an organism leads to alteration in lipid and carbohydrate metabolism. Lower concentration of PUFA in the skeletal muscles of human subjects has been associated with the condition of insulin resistance [5]. In rats with omega-3 deficiency, dysfunctions of insulin receptor signaling have also been observed, resulting in increased susceptibility to a high fructose challenge [6]. While the effects of dietary DHA supplementation have been extensively studied in mammals, the physiological role of endogenously de novo synthetized PUFA is not clear. It has been postulated that interference with the enzymatic machinery necessary for omega-3 and omega-6 fatty acids synthesis would have effects on systemic metabolism [7,8,9]. We have previously demonstrated that ablation of the PUFA elongase Elongation of very long-chain fatty acids 2 (ELOVL2), which is an essential enzyme in the endogenous production of DHA [10,11], leads to systemic depletion of this essential fatty acid in mice, and seems to be important factor in the maintenance of energy homeostasis [12]. Elovl2 -/- animals are resistant to diet-induced obesity, and administration of dietary DHA in those mice resulted in fully recovered DHA levels as well as weight gain. It has been proposed that the effects seen upon DHA-supplementation are influenced by the dose and duration of the treatment but also of the fatty acid source as well as of particular macronutrients in the background diet [13]. High sucrose high fat diet treatment in mice is frequently used as a common stressor to develop diet-induced obesity and consequently impaired fat and glucose metabolism [14]. In a previous study, we used diets, including the DHA-enriched diet, that were supplemented with high levels of sucrose as a second source of carbohydrates after starch (22% by energy). The aim of the present study was to investigate if the amount of sucrose in the DHA-supplementation diet influences lipid and glucose metabolism as well as weight gain in DHA-deficient Elovl2 -/- animals.

## 2. Material and Methods

### 2.1. Animal Handling and Experimental Set Up

Elovl2 -/- mice, backcrossed on 129S2/Sv for four generations [12] were housed at RT (22 °C) and maintained on 12 hours light/dark cycle and fed ad libitum and had free access to water. At the start of the experiment, Elovl2 wild-type and Elovl2 -/- mice were single caged and challenged with five different diet regimes consisting of 1) low-sucrose (LS) pre-treatment for 4 weeks prior to 4 weeks of high-sucrose, high-fat diet (HSHF), 2) low-sucrose diet plus DHA (LSDHA) for 4 weeks prior to 4 weeks of HSHF diet, 3) high-sucrose diet (HS) for 4 weeks prior to 4 weeks of HSHF diet, 4) high-sucrose diet plus DHA (HSDHA) for 4 weeks prior to 4 weeks of HSHF diet and 5) high-sucrose diet pre-treatment (HS) plus additional 4 weeks with HS. For the composition of diets, see Table 1 and for the experimental setup, see Figure 1. Age-matched littermates from the heterozygote breeding were used as controls and housed under the same conditions as the Elovl2 -/- mice. Body weight and food consumption were measured weekly. At the end of the study, animals were sacrificed with CO_2_. Relevant tissues were harvested and frozen in liquid nitrogen and stored at −80 °C for further analysis. All studies were carried out with ethical permission from the Animal Ethics Committee of the North Stockholm region, Sweden.

### 2.2. MRI Measurements

In vivo magnetic resonance imaging (MRI) using EchoMRI-100^TM^ (Echo medical Systems, Houston, Texas) was performed to measure fat mass and lean mass at the beginning of the experiment, after four initial weeks of treatment and at the end point.

### 2.3. Quantitative RT-PCR

Total RNA was isolated from hypothalamus and white epididymal adipose tissue with TRIReagent (Sigma Aldrich, St. Louis, MO, USA) following the manufacturer’s procedure. For real time PCR, 500 ng total RNA was reverse transcribed using random hexamer primers, dNTPs, multiscript and RNase inhibitor (Applied Biosystems, Foster City, CA, USA). cDNA samples were diluted 1:10 and aliquots of 2 μL of the sample cDNA were mixed with SYBR Green JumpStart TaqReadyMix (Sigma Aldrich, St. Louis, MO, USA), pre-validated primers, DEPC-treated water and were measured in duplicate for each sample. PCR-primers were used for CART, NPY, AgRP, POMC, PPARγ, Elovl3, Glut4, Cpt1, leptin, and adiponectin. For primer sequences, see Appendix A. Expression analysis was performed using BioRad Thermocycler. Data were normalized to the housekeeping gene 18S or TFIIB.

### 2.4. Glucose Tolerance Tests

Single caged Elovl2 -/- mice and wild-type littermates were analyzed at the starting point of the experiment as well as after four weeks of pretreatment and two weeks of high-fat diet treatment (in total, six weeks’ treatment) (Appendix A). The animals were starved for 6 hours starting at 09:00 prior to an intraperitoneal injection 2 g glucose per kg of body weight. Before injection, as well as 15, 30, 60, and 90 min post injection, glucose levels were measured using a MultiCare-in (Pedihealth, Finland) diagnostic device.

### 2.5. Blood TG and Cholesterol Levels

After performance of the glucose tolerance test at two weeks of high-fat diet treatment, triglyceride and cholesterol levels were measured using a MultiCare-in (Pedihealth, Finland) diagnostic device.

### 2.6. Serum/Plasma Leptin and Insulin Levels

Analysis of serum leptin and insulin levels was performed at the end point of the experiment using mouse serum leptin (Mouse Leptin, Enzo LifeScience) and insulin (Ultra-Sensitive Mouse Insulin, Crystal Chem Inc) ELISA kits.

### 2.7. Serum Lipoprotein Profile (FPLC)

To analyze the triglyceride and cholesterol content of the different classes of plasma lipoproteins, FPLC analyses were performed. Pooled plasma samples were separated on S6-superose columns (GE Healthcare) and lipid (TG and cholesterol) levels were analyzed in the different fractions using commercially available kits (Roche).

### 2.8. Intestinal Lipid Uptake Test

To investigate the rate of chylomicron production in the intestine, mice were injected intravenously with 0.5 mg/g BW Triton WR-1339 (Tyloxapol (Sigma); 10% solution in PBS) before receiving an oral gavage of olive oil containing 14C-Triolein (Perkin-Elmer; 1.5 kBq 14C-Triolein/g BW in 2.6 µL olive oil/g BW). Blood samples were taken from the tail vein before gavage and 30, 60, 120, and 240 min after gavage. Plasma samples were mixed with scintillation fluid and radioactivity was measured using a Perkin Elmer Tricarb scintillation counter and expressed as cpm per µL plasma.

### 2.9. Statistical Analysis

Data were analyzed by means of Prism 4 software (GraphPad Software, San Diego, CA). Differences between two or multiple groups were analyzed by Student’s t-test or ANOVA followed by Bonferroni post hoc test. A *p* value < 0.05 was considered significant. Data were expressed as mean ± SEM (standard error of the mean).

## 3. Results

### 3.1. DHA Supplementation Together with High Sucrose Content Induces Body Weight Gain of Elovl2 -/- Mice

To determine the role of sucrose on the effect of DHA-supplementation on body weight gain, we have fed wild-type and Elovl2 -/- mice with five different diet regimes (Figure 1). Animals pre-fed low-sucrose DHA-enriched diet (LSDHA), did not show significant changes in body weight during the five-weeks treatment (Figure 2A). After four weeks of HSHF challenge, we observed a significant body weight gain in wild-type animals pre-treated with LS diet, which was abolished by DHA supplementation (LSDHA), and not seen in Elovl2 -/- mice on either feeding regime (Figure 2A). In contrast, Elovl2 -/- mice pre-treated with the HSDHA diet and then fed HSHF diet gained significantly more weight compared to the wild-type mice under the same condition (Figure 2B). Weight gain in this group was also higher than in both wild-type and Elovl2 -/- mice pre-treated with only HS prior to four weeks of HSHF (Figure 2B). Interestingly, the weight gain of the Elovl2 -/- mice was already pronounced during the four weeks HSDHA pre-treatment period before the mice were exposed to a high-fat diet, highlighting the profound effects of the presence of DHA in mice that never experienced DHA exposure before.

Body composition analysis at the beginning of the study did not show any significant variation in fat and lean mass in both genotypes (Table 2 and Figure 2C,D). However, fat and lean mass measured after four weeks of pre-treatment and at the end of the experiment revealed that the changes in body weight were mainly reflected by a change in fat deposition (Figure 2C,D). In addition, both the control groups of wild-type and Elovl2 -/- mice fed HS for eight weeks showed an increase in fat accumulation, although Elovl2 -/- mice tended to store less fat than their wild-type littermates (Table 2). The effects of the sucrose content of the diet on weight gain led us to speculate whether the different dietary composition would influence food intake or appetite.

### 3.2. Effects of High Sucrose and DHA Supplementation on Energy Intake

The effect on body weight gain after four weeks of HSHF diet could be explained by an increased food consumption in these mice. However, even though all experimental groups showed a temporary increase in food intake during the first week of HSHF diet, there was no significant difference in energy intake (Figure 3A,C) or cumulative food consumption (Figure 3B,D).

Leptin is an important regulator of appetite and is secreted by white adipose tissue in proportion to its lipid storage. We did not detect any significant differences in the concentration of leptin in wild-type and Elovl2 -/- animals in any experimental group (Appendix A) arguing against an effect of the sucrose content of the diet on appetite. However, as expected, the high-fat diet treatment increased leptin levels as compared to control groups treated with high-sucrose but low-fat diet (Appendix A), in line with the general view that increased adipose mass leads to increased leptin secretion. Additionally, and in line with the unaltered food intake and leptin levels, analysis of the transcription of orexigenic peptides NPY and AGRP and the anorexigenic peptides POMC and CART in the hypothalamus of wild-type and Elovl2 -/- mice revealed no alteration in mRNA levels in response to HS or HSDHA diet followed by HSHF diet (Figure 4). Thus, the body weight gain in the HS pre-treatment group cannot be explained by a regulation of the central food intake controlling mechanism in the hypothalamus. We therefore wondered if changes in nutrient availability, rather than food intake, could drive the differences in weight gain.

### 3.3. DHA Supplementation and High-Sucrose Content Improves Food Utilization Efficiency

Metabolic efficiency, expressed as percent of food stored as fat, was calculated based on the energy intake and fat mass data for all experimental groups after four weeks of pre-treatment (LS, LSDHA, HS, HSDHA) and after four weeks of HSHF diet. During the pre-treatment period, DHA supplementation with low sucrose (LSDHA) slightly increased food utilization in both wild-type and Elovl2 -/- mice as compared to their littermates kept on LS diet (Figure 5A). Pre-treatment with HS diet increased metabolic efficiency in both genotypes (Figure 5B), pointing towards increased lipogenesis and fat accumulation under these conditions of high sucrose availability. When DHA was supplemented together with high-sucrose diet (HSDHA), Elovl2 -/- mice displayed a more efficient deposition of energy as compared to the DHA-deficient mice fed only HS diet (Figure 5B), reflecting the changes in body weight observed under these conditions (Figure 2B). A positive effect of sucrose on metabolic efficiency was also observed in both genotypes of the control groups maintained on HS diet for eight weeks (Table 2), while the Elovl2 -/- mice showed a lower metabolic efficiency than their wild-type littermates. Four weeks of HSHF diet feeding significantly increased the metabolic efficiency, especially in wild-type mice pretreated with LS diet (Figure 5C), which is in line with the observed differences in body weight gain (Figure 2A). There was no difference in the metabolic efficiency between the experimental groups upon HSHF treated after four weeks of HS pretreatment (Figure 5D).

To exclude the possibility that food utilization reflects altered food digestibility and absorption, we measured feces mass and calculated apparent absorbed food mass during digestion. Irrespective of the type of diet, as well as genotype, we did not observe any significant difference for the calculated efficiency of mass absorption (Appendix A). To investigate intestinal lipid uptake more closely, we performed an intestinal chylomicron production test in wild-type and Elovl2 -/- mice maintained on low-sucrose (LS) diet. Gavage with olive oil spiked with ^14^C-triolein demonstrated similar recovery of radioactivity in the circulation after 30, 60, 90, 120, and 240 min in both genotypes (Appendix A), indicating unaltered intestinal lipid absorption. Taken together, these results show that the differences observed in weight gain are mainly explainable by changes in the metabolic efficiency, without food intake, appetite, or absorptive capacity being altered.

### 3.4. Effects of High Sucrose and DHA Supplementation on Glucose Metabolism

Increased adiposity has been repeatedly shown to be a risk factor for the development of insulin resistance and hyperglycemia in mice and humans. We thus wondered if the observed changes in body fat accumulation would translate into altered systemic metabolism and glucose homeostasis. The impact of endogenous DHA alteration on carbohydrate metabolism was analyzed in glucose tolerance tests performed before the introduction of the specific diet regimes, and after two weeks of HSHF treatment (at week 6 of experiment) in animals pre-treated with HS or HSDHA diet (Figure 1). At the starting point, when animals were fed regular low-sucrose diet (LS), the response to glucose was the same in Elovl2 -/- mice and wild-type animals as glucose levels returned to initial fasting levels within 60 min (Figure 6A). Similarly, the Elovl2 -/- mice pre-treated with HS and HSDHA diet, followed by two weeks of HSHF diet, responded to the glucose challenge in the same way as their wild-type littermates, and their initial fasting levels of glucose were regained 90 min post glucose injection (Figure 6B).

However, Elovl2 -/- animals fed the HS diet (low fat) for eight weeks tended to be more glucose tolerant than wild-type mice and showed lower levels of glucose already after 30 min after the glucose administration (Figure 6C). The analysis of serum insulin levels in the fed stage of animals, performed at the end of experiment, did not show any significant difference between the experimental groups (Figure 6D). There was no significant difference in fasting glucose levels between Elovl2 -/- and wild-type mice maintained on LS diet at the start of the experiment (Appendix A). However, after four weeks of LSDHA diet, the Elovl2 -/- mice showed a tendency towards reduced glucose levels compared with wild-type littermates (Appendix A). After an additional four weeks of HSHF diet, the glucose levels were lower in the Elovl2 -/- mice pretreated with DHA than in wild-type mice and significantly lower than in Elovl2 -/- mice that had not been supplemented with DHA (Appendix A), despite increased body weight in these animals.

### 3.5. Effects of DHA and Sucrose on Lipid Metabolism Markers

Since there was an increased accumulation of fat in the Elovl2 -/- mice upon high-sucrose DHA-enriched diet, we analyzed the expression of certain genes involved in adipogenesis in epididymal white adipose tissue harvested from animals fed HS diet for the whole eight-week period and from mice pre-treated with HSDHA or LSDHA for four weeks prior to four weeks of HSHF diet. As seen in Figure 7, the transcription of Peroxisome-proliferator-activated-receptor-gamma (PPARγ) and its target genes such as Glucose transporter 4 (GLUT4) and elongation of very long-chain 3 (Elovl3) were elevated in Elovl2 -/- animals compared with wild-type littermates maintained on HS diet (Figure 7A–C). Interestingly, this expression pattern did not follow the same trend when animals were exposed to DHA supplementation prior to HSHF diet (Figure 7A–C). The mRNA levels of the fatty acid oxidation marker carnitine palmitoyltransferase 1 (Cpt1), did not differ between the genotypes but showed a tendency to decrease in response to HSHF diet treatment upon DHA pretreatment implying increased fatty acid utilization in Elovl2 -/- mice (Figure 7D). In accordance with the fat mass data, leptin gene expression as well as serum levels, were increased in both wild-type and Elovl2 -/- animals on HSHF diet pretreated with HSDHA, but not in Elovl2 -/- mice pre-treated with LSDHA diet (Figure 7E and Appendix A), supporting the idea that DHA supplementation in conjunction with sucrose feeding increases fat deposition in Elovl2 -/- mice. In contrast, the expression of adiponectin in white adipose tissue of Elovl2 -/- mice did not differ significantly in any of the experimental groups (Figure 7F). Moreover, triglyceride (TG) concentration in the blood of Elovl2 -/- mice maintained on HS diet followed by HSHF was lower (1.5 ± 0.1 mM) than in the wild-type littermates (1.7 ± 0.2 mM) independently of DHA supplementation (Appendix A). In line with the TG concentration in the blood, lipoprotein profile of serum triglycerides confirmed a lower number of TG-rich lipoproteins (TRL) in Elovl2 -/- mice kept on HS diet followed by HSHF treatment. However, when DHA-deficient Elovl2 -/- animals were exposed to HSDHA and subsequently HSHF, the TRL fraction was re-established to the same concentration level as for wild-type animals (Figure 8A). Regarding the total cholesterol blood concentration and serum cholesterol lipoprotein profile, we did not observe any differences between wild-type (5.1 ± 0.2 mM) and Elovl2 -/- animals (4.7 ± 0.3 mM) pre-treated HS or HSDHA (4.7 ± 0.1 mM and 4.8 ± 0.2 mM, respectively) (Figure 8B and Appendix A).

## 4. Discussion

Our previous findings have shown that Elovl2 -/- mice maintained on high-fat diet for four weeks did not gain body weight, which was mainly due to lower fat accretion in the DHA-deficient Elovl2 -/- mice in comparison with their wild type littermates. A two-week pre-treatment of DHA-supplemented diet prior to the high fat exposure was able to restore systemic levels of DHA to normal, which also abolished the resistance to diet-induced obesity in the Elovl2 -/- mice [12]. These results are somewhat surprising since DHA is rather accepted as a beneficial factor for treatment of obesity and its related diseases [15,16,17]. To understand the mechanism behind the positive effect of DHA supplementation on fat accretion in the Elovl2 -/- mice, here we have focused on the combined effect of DHA and sucrose in the diet. In previous experiments, we used a DHA-supplemented diet that was rich in sucrose. In the present investigation, we show that Elovl2 -/- mice supplemented with a low-sucrose DHA-enriched diet, in contrast to a high-sucrose DHA-enriched diet, did not gain weight as a result of less fat accretion. In agreement with earlier results, the weight gain and the fat accumulation seen in the DHA-sufficient wild-type mice exposed to four weeks high fat/high-sucrose diet were abrogated by DHA supplementation (Figure 9). However, in contrast to the Elovl2 -/- mice, supplementation of DHA combined with sucrose did not have any effect on weight gain of the wild-type mice. This clearly shows that sucrose is an important factor modulating the effects of DHA supplementation. An imbalance in cellular DHA concentration has been associated with changes in membrane properties that are of major significance for proper neuronal satiety signaling, including the release appetite mediators such as orexigenic and anorexigenic neuropeptides and in hypothalamus, affecting the central control of food consumption [18,19,20,21]. Although there was a tendency for both wild-type and Elovl2 -/- mice to eat more after the shift from a low-sucrose diet to a high-sucrose, high-fat diet, we did not observe any significant difference in the weekly and cumulative energy intake between the different genotype and treatments. In addition, the expression of the hypothalamic neuropeptides involved in the control of food intake was not significantly different between Elovl2 -/- and wild-type littermates and was not affected by DHA supplementation. Deficiency in PUFA can be linked to inflammation and problems with food digestibility and assimilation [9,22]. To exclude any perturbation in nutrient absorption, the DHA-deficient Elovl2 -/- mice did not show any significant impairment in lipid recovery when compared with wild-type littermates. However, to fully exclude impaired food digestibility in the DHA-deficient mice, analysis of the energetic content of the feces is required. Analysis of Fads2 -/- mice (another endogenous PUFA deprived model system) also revealed an obesity resistance phenotype that, upon DHA treatment, resulted in hyperphagia together with increased body weight as well as enlarged white adipose tissue mass and size of adipocytes. However, the authors did not state any information about the sucrose content in the diet used in their study [9]. The alteration in metabolic efficiency in the Elovl2 -/- mice, which is not an effect of altered energy intake, suggests an alteration in energy expenditure in these animals. We have earlier shown that Elovl2 -/- mice maintained on low-sucrose diet show a lower respiratory quotient (RQ) value during the dark phase, implying a higher preference for lipid utilization in a condition of PUFA deficiency [12]. However, in that study, we did not record any significant difference in total energy expenditure between the Elovl2 -/- mice and corresponding wild-type littermates which could explain the impaired fat mass in these animals. Sucrose has previously been postulated as being a factor abrogating the anti-obesity effects of omega-3 PUFA enriched diets [13,23,24]. The adverse effect of combined PUFA and sucrose treatment has been related to increased insulin secretion and insulin mediated glucose uptake [24]. Moreover, hyperinsulinemic *ob/ob* animals treated with fish oil showed increased weight gain [25]. In addition, rats fed a diet rich in sucrose and linoleic acid develop hyperglycemia and hyperinsulinemia, an effect suggested to be attributed by accumulation of arachidonic acid, which was also observed in DHA-deficient mice [12], and the insulinotropic effect of linoleic acid [26]. Feeding Sprague-Dawley rats with an omega-3-deficient and high fructose diet influenced glucose uptake and TG levels due to deficient insulin signaling [6]. Therefore, altered fatty acid composition together with a diet high in carbohydrates is suggested to result in a higher vulnerability towards metabolic challenge [27]. Recently, a role of ELOVL2 in ensuring normal insulin secretory responses to glucose [28], as well as a role of ELOVL2 in pancreatic beta-cell function [29] have been suggested. Although DHA-deficiency in our Elovl2 -/- mice fed standard chow diet (low in sucrose and DHA) did not lead to any alterations in glucose tolerance or insulin levels (in the fed stage), we cannot rule out an altered insulin signaling upon DHA-sucrose supplementation in these animals. In addition, lower levels of DHA in the brain of rats maintained on omega-3 deficient diet showed reduced glucose utilization together with lower expression of brain glucose transporters [30,31]. Therefore, to obtain comprehensive information about glucose metabolism in DHA-deficient animals, we suggest further studies on in vivo glucose uptake in the brain and muscles of e.g., Elovl2 -/- mice. Moreover, as fatty acids secreted from adipose tissue and liver are suggested to be important mediators in the maintenance of energy homeostasis, the relative strong alteration in the transcription of the PPARγ target gene, *Elovl3*, involved in the synthesis of mono-unsaturated fatty acids known to be important for lipid accretion, in the Elovl2 -/- mice should be taken in consideration for further studies [32,33]. The link between DHA and TG formation is also supported by *Elovl2* overexpression in an adipocyte cell line, where higher levels of ELOVL2 resulted in increased expression of PPARγ target genes, together with elevated TG formation and lipid accumulation [34]. From the lipoprotein profile in the Elovl2 -/- mice it can be concluded that DHA supplementation together with sucrose improves the serum level of TG-rich lipoproteins to the levels seen in wild-type mice. Especially in light of the recent discovery of the role of DHA in the prevention and treatment of adipose tissue inflammation [35,36], both endogenously formed as well as dietarily supplied DHA are therefore suggested to control white adipose tissue metabolism and energy balance in a multifactorial way which needs further insights.

## Figures and Tables

**Figure 1 nutrients-11-00852-f001:**
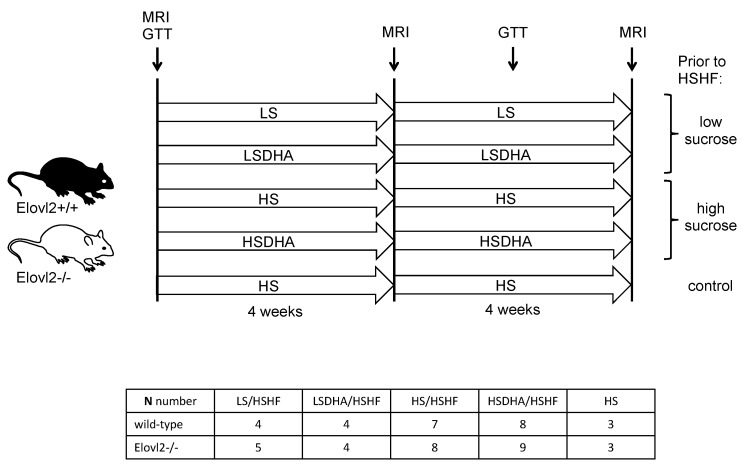
Experimental set-up. Wild-type and Elovl2 -/- mice were kept on low-sucrose (LS) or low-sucrose DHA-enriched diet (LSDHA) or on high-sucrose (HS) or high-sucrose DHA-enriched diet (HSDHA) for four weeks, followed by four weeks of high-sucrose, high-fat diet (HSHF) feeding. Control groups were kept on high-sucrose diet (HS) for eight weeks. Magnetic resonance imaging (MRI) was performed to measure fat and lean mass at the start point of the experiment, after four weeks pre-treatment and at the end of the experiment after four weeks of high-sucrose, high-fat diet. Glucose tolerance tests (GTT) were performed at the starting point and after six weeks of treatment (four weeks of pre-treatment and two weeks of high-sucrose, high-fat diet (HSHF).

**Figure 2 nutrients-11-00852-f002:**
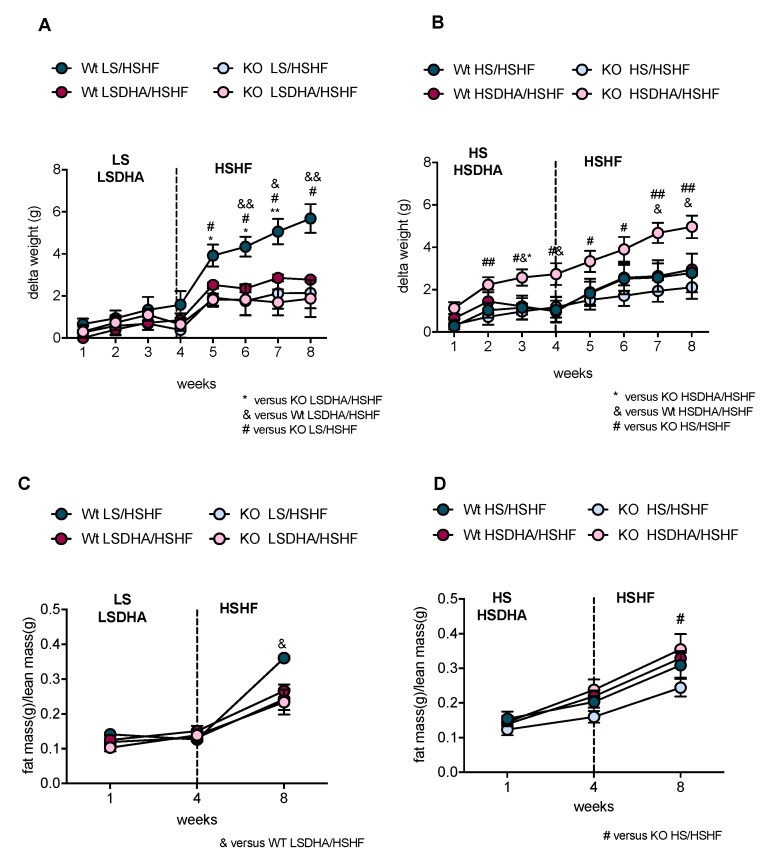
Sucrose content in DHA-enriched diet has an impact on gaining of weight by Elovl2 -/- mice. (**A**) Development of body weight in wild-type and Elovl2 -/- animals fed low-sucrose diet (LS) or low-sucrose DHA-enriched diet (LSDHA), followed by high-sucrose, high-fat diet (HSHF). (**B**) Body weight of animals fed high-sucrose (HS) or high-sucrose DHA-enriched diet (HSDHA) followed by high-sucrose, high fat diet (HSHF). (**C**) Obesity index for wild-type and Elovl2 -/- mice, presented as ratio of fat mass (g) to lean mass (g), at the beginning of the treatment, after four weeks of low-sucrose (LS) or low-sucrose DHA-enriched diet (LSDHA), and after additional four weeks of HSHF diet. (**D**) Obesity index at the beginning of the treatment, after four weeks of high-sucrose (HS) or high-sucrose DHA-enriched diet (HSDHA), and after four weeks of high-sucrose, high-fat diet (HSHF). Results shown are means of 4–9 mice ± SEM. Statistical significances are shown between groups * *p* < 0.05 and ** *p* < 0.01. ^#^
*p* < 0.05 and ^##^
*p* < 0.01. ^&^
*p* < 0.05 and ^&&^
*p* < 0.01.

**Figure 3 nutrients-11-00852-f003:**
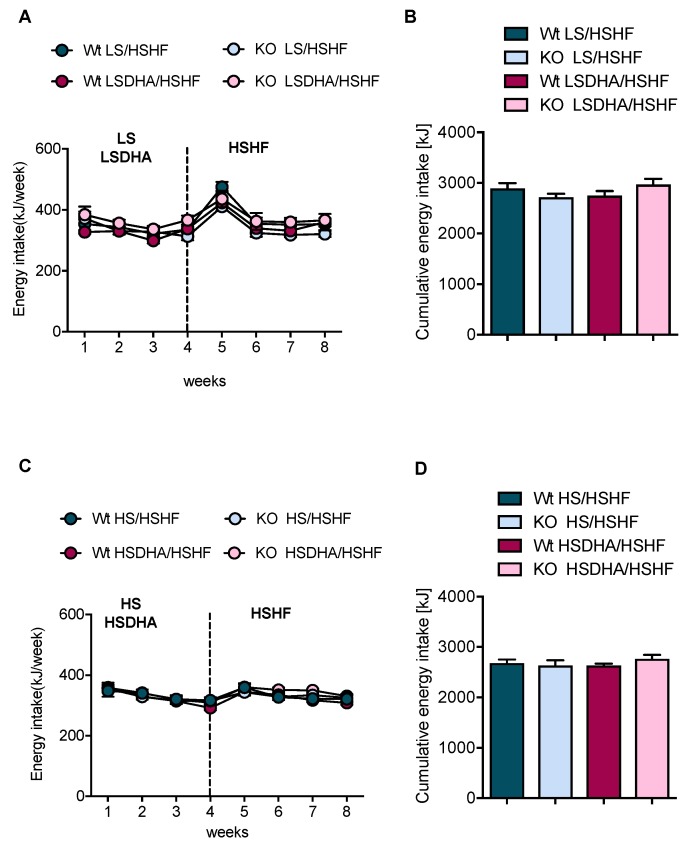
DHA supplementation and sucrose content do not affect energy intake. (**A**) Weekly energy consumption of wild-type and Elovl2 -/- mice fed low-sucrose diet (LS) or low-sucrose DHA-enriched diet (LSDHA), followed by high-sucrose, high-fat diet (SHF). (**B**) Cumulative energy intake, presented as kJ intake per eight weeks, for wild-type and *Elovl2-/-*mice fed low-sucrose diet (LS) or low-sucrose DHA-enriched diet (LSDHA) followed by high-sucrose, high-fat diet (HSHF). (**C**) Weekly consumption of animals fed high-sucrose (HS) or high-sucrose DHA-enriched diet (HSDHA), followed by high-sucrose, high-fat diet (HSHF). (**D**) Cumulative energy intake of animals fed high-sucrose (HS) or high-sucrose DHA-enriched diet (HSDHA) followed by high-sucrose, high-fat diet (HSHF). Results shown are means of 4–9 mice ± SEM.

**Figure 4 nutrients-11-00852-f004:**
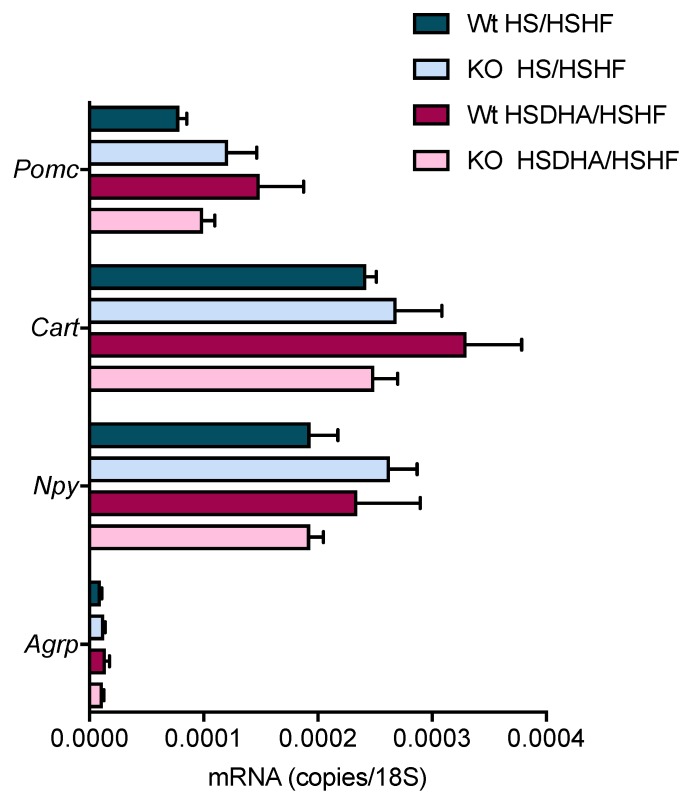
Hypothalamic mRNA expression of neuropeptides involved in the control of food intake is not affected by supplementation of DHA with high sucrose content. Relative hypothalamic gene expression of the orexigenic peptides NPY and AGRP and the anorexigenic peptides POMC and CART in wild-type and Elovl2 -/- mice fed high-sucrose (HS) or high-sucrose DHA-enriched diet (HSDHA) followed by high-sucrose, high-fat diet (HSHF) feeding. mRNA levels are shown relative to 18S expression. Results shown are means of four mice ± SEM.

**Figure 5 nutrients-11-00852-f005:**
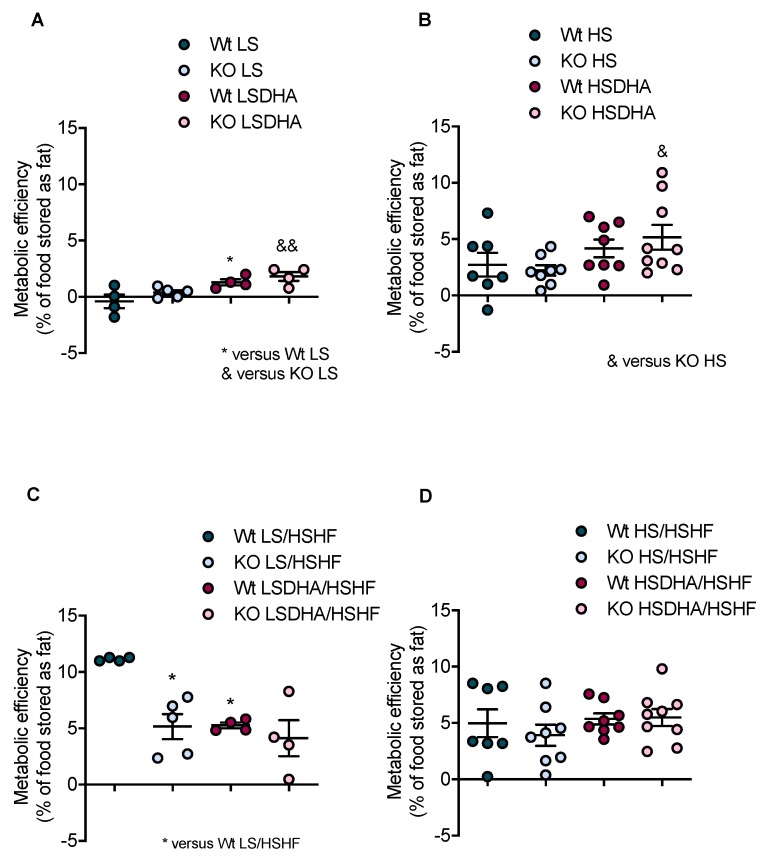
High dietary sucrose content and DHA supplementation improves metabolic efficiency of Elovl2 -/- mice. (**A**) Metabolic efficiency, presented as % food stored as fat, in wild-type and Elovl2 -/- mice after feeding low-sucrose diet (LS) or low-sucrose DHA-enriched diet (LSDHA). (**B**) Metabolic efficiency after feeding high-sucrose (HS) or high-sucrose DHA-enriched diet (HSDHA). (**C**) Metabolic efficiency after feeding of high-sucrose, high-fat diet (HSHF) in animals pre-treated with low-sucrose diet (LS) or low-sucrose DHA-enriched diet (LSDHA). (**D**) Metabolic efficiency after feeding of high-sucrose, high-fat diet (HSHF) in animals pre-treated with high-sucrose (HS) or high-sucrose DHA-enriched diet (HSDHA). Results shown are means of 4–9 mice ± SEM. ^&^
*p* < 0.05 and ^&&^
*p* < 0.01. * *p* < 0.05.

**Figure 6 nutrients-11-00852-f006:**
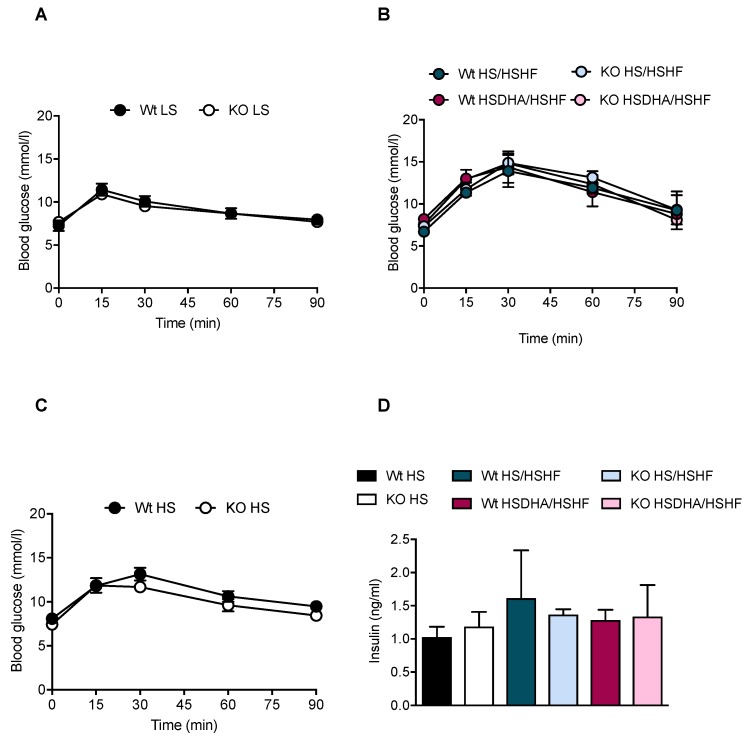
Glucose tolerance of Elovl2 -/- mice is not affected by high dietary sucrose content and DHA supplementation. (**A**) Glucose tolerance test performed on wild-type and Elovl2 -/- mice fed standard chow diet at the start point of experiment. (**B**) Glucose tolerance test after six weeks of dietary treatment (four weeks of high-sucrose diet (HS) or high-sucrose DHA-enriched diet (HSDHA) followed by two weeks of high-sucrose, high-fat (HSHF)). (**C**) Glucose tolerance test performed on wild-type and Elovl2 -/- mice fed high-sucrose diet for eight weeks. (**D**) Insulin levels after GTT of the groups shown in **B**,**C**. Results shown are means of four mice ± SEM.

**Figure 7 nutrients-11-00852-f007:**
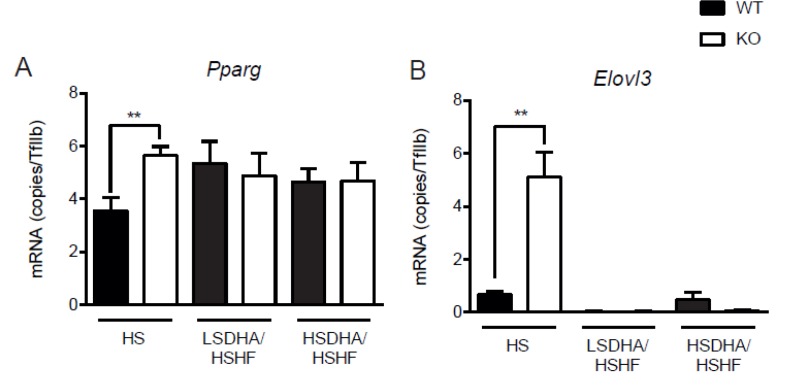
Dietary sucrose content affects expression of genes involved in the accumulation of fat in the epididymal white adipose tissue of Elovl2 -/- mice. Relative epididymal white adipose tissue gene expression of PPARγ (**A**), Elovl3 (**B**), Glut4 (**C**), Cpt1 (**D**), leptin (**E**), and adiponectin (**F**) in wild-type and Elovl2 -/- mice fed low-sucrose DHA-enriched diet, followed by high-sucrose, high-fat diet (LSDHA/HSHF); high-sucrose DHA-enriched diet followed by high-sucrose, high-fat diet (HSDHA/HSHF) or mice that were just fed high-sucrose diet (HS). mRNA levels relative to TFIIB expression. Results shown are means of ±SEM of four mice. * *p* < 0.05 and ** *p* < 0.01 versus WT by paired Student’s t-test.

**Figure 8 nutrients-11-00852-f008:**
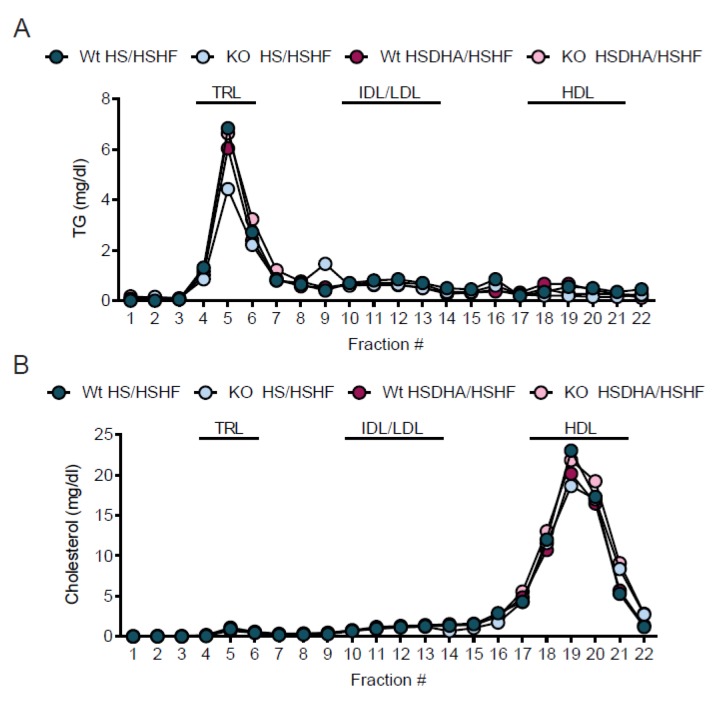
Triglyceride lipoprotein profile in Elovl2 -/- mice is reversed to wild-type levels after high-sucrose DHA-enriched (HSDHA) treatment. Serum lipoprotein triglyceride (**A**) and cholesterol (**B**) profiles were determined on pooled serum from four to five mice per group of wild-type and Elovl2 -/- mice fed high-sucrose (HS) or high-sucrose DHA-enriched diet (HSDHA) followed by high-sucrose, high-fat (HSHF) diet. Fractions 4–6 represent TG-rich lipoproteins (TRL), fractions 10–15 low density lipoprotein and mainly remnant particles (LDL), and fractions 16–23 represent high density lipoproteins (HDL).

**Figure 9 nutrients-11-00852-f009:**
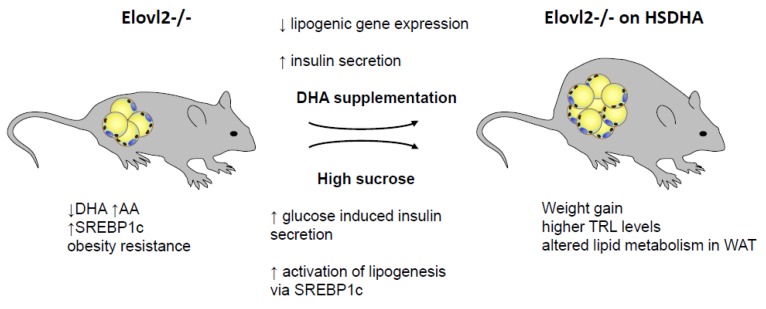
High sucrose together with DHA supplementation affect lipid metabolism in Elovl2 -/- mice are characterized by low levels of DHA and resulting increases in the levels of arachidonic acid (AA). On an obesogenic diet, these animals show resistance to diet-induced obesity. These mice were fed a DHA-enriched diet (reducing lipogenic gene expression and at the same time increasing insulin secretion) containing high levels of sucrose (resembling human diet; also, leading to increased insulin levels and at the same time activating lipogenesis). This treatment of Elovl2 -/- mice resulted in a loss of the protection against diet-induced obesity, increased plasma TRL levels, and led to altered expression of key players in metabolic control in adipose tissue.

**Table 1 nutrients-11-00852-t001:** Composition of diets. Ingredients presented as gram of total mass.

Ingredients	No Sucrose Diet (LS) (10% kcal fat) D12450B	No Sucrose DHA-Enriched Diet (LSDHA) (10% kcal fat, 1% DHA) D14062901	High Sucrose Diet (HS) (10% kcal fat) D12450H	High Sucrose DHA-Enriched Diet (hsdha) (10% kcal fat, 1% DHA) D13021002	High Sucrose, High Fat Diet (HSHF) (45% kcal fat) D12451
Casein	200	200	200	200	200
L-Cystine	3	3	3	3	3
Corn starch	315	550	452.2	452.2	72.8
Maltodextrin 10	35	150	75	75	100
**Sucrose**	**0**	**0**	**172.8**	**172.8**	**172.8**
Cellulose, BW200	50	50	50	50	50
Soybean oil	25	25	25	25	25
**DHA**	**0**	**10**	**0**	**10**	**0**
Lard	20	10	20	10	177.5
Mineral Mix S10026	10	10	10	10	10
DiCalcium Phosphate	13	13	13	13	13
Calcium Carbonate	5.5	5.5	5.5	5.5	5.5
Potassium Citrate, 1H_2_0	16.5	16.5	16.5	16.5	16.5
Vitamin Mix V10001	10	10	10	10	10
Choline Bitartrate	2	2	2	2	2
FD&C Yellow Dye #5	0	0	0.04	0	0
FD&C Red Dye #40	0	0.05	0.01	0.025	0.05
FD&C Blue Dye #1	0	0	0	0.025	0
Total (g)	705	1055.5	1055.5	1055.05	858.15

Diet composition, expressed in gram of mass for each ingredient, of chow diet with low sucrose content (LS) (Labfor R70, Lantmännen, Sweden, less than 10% kcal fat), low-sucrose DHA enriched diet (LSDHA) (10% kcal fat, 1% DHA, D14062901, Research Diets, New Brunswick, NJ, USA), high-sucrose diet (HS) (10% kcal fat, D12450H, Research Diets, New Brunswick, NJ, USA), high-sucrose DHA enriched diet (HSDHA) (10% kcal fat, 1% DHA, D13023002, Research Diets, New Brunswick, NJ, USA), and high-sucrose, high-fat diet (HSHF) (45% fat, D12451, Research Diets, New Brunswick, NJ, USA) formulated by Research Diets, New Brunswick, NJ, USA. FD&C (Food, Drug, and Cosmetic) were added in order to distinguish by color the different diets.

**Table 2 nutrients-11-00852-t002:** Parameters of Elovl2+/+ (Wt) and Elov2-/- (KO) mice on control HS diet (for 8 weeks).

**Initial Parameters**	**Wt HS (8 weeks)**	**KO HS (8 weeks)**
Body weight (g ± SEM)	29.9 ± 1.2	27.8 ± 1.6
Lean weight (g± SEM)	22.05 ±0.99	20.74 ± 1.34
Fat mass (g± SEM)	3.22 ± 0.14	2.75 ± 0.09
**Final Parameters**	**Wt HS (8 weeks)**	**KO HS (8 weeks)**
Body weight (g± SEM)	30.4 ± 0.7	27.2 ± 0.2
Lean weight (g± SEM)	20.29 ± 0.39	19.07 ± 0.74
Fat mass (g± SEM)	4.98 ± 0.37	3.52 ± 0.29
**Weekly Food Consumption (kJ/week ± SEM)**	**325.1± 6.4**	**313.7 ± 14.9**
Cumulative energy intake (kJ/8 weeks± SEM)	2600.9 ± 51.0	2509.5 ± 119.5
Metabolic efficiency (% of food store as fat)	2.5	1.1

Body weight, fat weight, lean weight at the start point of the experiment (initial) and at the end (final), weekly and cumulative energy intake, and metabolic efficiency expressed for wild-type and Elovl2 -/- mice fed for eight weeks high sucrose diet (HS). All results are mean ± SEM (standard error of the mean) and representative of three animals per group analyzed by paired Student’s t-test.

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
