# Peer review of "Synergistic Effects of DHA and Sucrose on Body Weight Gain in PUFA-Deficient Elovl2 -/- Mice"

_nutrients, 2019, doi:10.3390/nu11040852_

Round 1

Reviewer 1 Report

The authors investigated the role of DHA and sucrose on lipids metabolism using Elovl2 KO mice model and found the co-regulation of DHA and sucrose on body weight gain in PUFA-deficient Elovel2 KO mice. There are a few questions the authors should address using current experimental conditions to enhance the quality of the manuscript.

1: Does Elovl2 involve DHA and sucrose-mediated insulin sensitivity?

2: Does Elovl2 KO WAT have different lipolysis under DHA and sucrose diet?

3: Does Elovl2 KO WAT have defective ChREBP activity and targets gene expression?

4: Does Elovl2 regulate sucrose-mediated fatty liver?

5: Does Elovl2 regulate ChREBP and SREBP1 activity and targets genes expression in liver?

6: Does Elovl2 KO mice have different physical activity under DHA and sucrose diet?

Author Response

Response to Reviewer 1 Comments

The authors investigated the role of DHA and sucrose on lipids metabolism using Elovl2 KO mice model and found the co-regulation of DHA and sucrose on body weight gain in PUFA-deficient Elovel2 KO mice. There are a few questions the authors should address using current experimental conditions to enhance the quality of the manuscript.

1: Does Elovl2 involve DHA and sucrose-mediated insulin sensitivity?

It has been suggested that Elovl2 plays role in ensuring normal insulin secretory responses to glucose and that Elov2 expression positively correlates with glucose intolerance and insulin secretion (Molecular phenotyping of multiple mouse strains under metabolic challenge uncovers a role for Elovl2 in glucose-induced insulin secretion (Cruciani-Guglielmacci et al., 2017)). Moreover Elovl2/DHA axis has been identified as factor stimulating FA oxidation in order to protect beta cells against glucolipotoxicity (Protective role of the ELOVL2/docosahexaenoic acid axis in glucolipotoxicity-induced apoptosis in rodent beta cells and human islets. (Bellini et al., 2018)). In case of our Elovl2-/- model we could observe, from unpublished data on isolated pancreatic islets of Elovl2-/-, that deficiency in DHA leads to impaired insulin secretion in response to glucose stimulus. This beta cells function seems to be recovered in isolates from Elovl2-/- fed DHA enriched diet. However, glycemia of Elovl2-/- does not significantly differ from the Wt and insulin released during the glucose tolerance test is not changed. Elovl2 deficiency doesn’t impact glucose tolerance and insulin secretion in our mice under applied dietary conditions. 

2: Does Elovl2 KO WAT have different lipolysis under DHA and sucrose diet?

We did not investigate lipolysis deeply and under sucrose diet but we have seen an upregulation of ATGL gene expression in KO compare to Wt in WAT under HF +/- DHA diet. Several studies have shown that ATGL-deficiency in mice is associated with reduced lipolysis resulting in excessive fat deposition in virtually all tissues. This implicates that the ATGL mediated catabolism of TAG is required in essentially all cell types of the body.

In line with this, we also have found a significant increase of ATGL expression in the retina of Elovl2 -/- mice together with a significant downregulation of lipid droplets accumulation in KO eyes compare to Wt (data unpublished). 

3: Does Elovl2 KO WAThave defective ChREBP activity and targets gene expression?

We did not analyze the expression of ChREBP but we looked at some of its target genes (SCD1, FAS and Elovl6) in WAT from Elovl2 -/- and Wt. We did not find any significant differences between the two genotypes under different diets but KO mice under HF diet (DHA -) tends to increase lipogenic genes SCD1 and FAS.

.

4: Does Elovl2 regulate sucrose-mediated fatty liver?

In our previous paper (Pauter AM et al 2014) we have investigated fatty liver in Wt and Ko mice fed with chow and high fat diet. Under chow diet, despite upregulation in hepatic lipogenic gene expression (SREBP1, FAS and SCD1) Elovl2 -/- mice did not show accumulation of liver TG and the histological analysis did not show any signs of fatty liver. Interestingly upregulation of SREBP1c by high-fat diet results in decreased expression of lipogenic genes in both Wt and KO mice suggesting an independent mechanism in the control of lipogenesis under these conditions. Despite overexpression of SREBP1 by high fat diet, KO mice did not show any sign of fat storage and steatosis, which again support the observation that reduced level of DHA in these mice are protective against fatty liver.   

5: Does Elovl2 regulate ChREBP and SREBP1 activity and targets genes expression in liver?

As explained in point 4,deficiency in the endogenous synthesized DHA activates hepatic SREBP1 (protein and mRNA level) and stimulates transcription of lipogenic target genes such as FAS and SCD1 in liver.  We also analyzed hepatic expression of PPARg and some of its target genes. We could not detect any differences between KO and Wt mice fed on chow diet. However, under high fat conditions there was a significant reduction of PPARg and PCK1 mRNA levels in Elovl2 -/- mice. This implies that omega3 fatty acids in addition to affecting lipogenesis through SREBP-1c activation can also modulate lipid homeostasis via SREBP1-indipendet pathway affecting lipid storage and possibly involving PPARg and pyruvate cycling.

6: Does Elovl2 KO mice have different physical activity under DHA and sucrose diet?

We did not notice any different physical activity between Wt and Ko mice under different diet regimes but from the measurement of resting metabolic rate we observed that Elovl2 -/- under chow diet have higher energy expenditure then Wt mice in the light period and at the beginning of dark period. Analysis of calorimetric parameter RQ, show significant lower level for Elovl2 -/- during the dark period when animals are more active and have more food intake (Pauter A.M at el. 2014). 

Furthermore, in data unpublished in order to see whether reduced expression of genes associated to neural plasticity and inflammation in Elovl2 -/- mice brain might have a physiological impact in this mouse model, we performed some behavioral analyses. Our results showed that none of the behavioral tests performed highlighted significant differences between the two genotypes.

Reviewer 2 Report

The manuscript by Anna M. Pauter et al. evaluated the effects of DHA and Sucrose on body weight gain in Elvol2-/- mice. The present study shows the role of Elovl2, DHA and sucrose on body weight gain, Metabolic efficiency, glucose, and lipid metabolism very elegantly. Potentially, the experiments are straightforward and nicely executed, and the manuscript is well written. Through this study and from their previous studies on Elovl2, have clearly shown the importance of DHA and role of Elovl2 in maintaining the proper lipid profile.  

The introduction and the discussion suggest some potential insights on the mechanisms by which Elovl2 contributes to body weight gain, glucose metabolism, lipid metabolism, and insulin resistance, but none were investigated in this study. The main focus was on the DHA and how supplementation restores the lipid profile.

It would be interesting to know the heart function of these mice, as a change in diet and supplementation affect the heart function. It would be an interesting study if the author can provide some data regarding heart function and blood pressure.   

Author Response

DHA supplementation is consider to be an important factor for the maintenance of cardiovascular system. We agree that our model organism, lacking endogenous DHA, is an interesting animal for studies on the heart function and blood pressure. From previously published whole genome studies we know that there is no association between genetic variation of elongases and intermediate cardiovascular phenotype or myocardial infarction (Genetic variation in fatty acid elongases is not associated with intermediate cardiovascular phenotypes or myocardial infarction (Aslibekyan et al., 2012)). We looked into the fatty acid composition of Elovl2-/- heart (left ventricle) maintained on DHA+/- diet. We could observe, as in the previously published on the systemic level, massive decreased in DHA level accompanied with increased of AA and EPA under DHA- diet. Moreover Elovl2-/- on DHA – diet showed tendency for the accumulation of TG and higher weight of the heart normalized to the body size. Described parameters (FA composition, TG content and heart weight) have not been changed in the Elovl2-/- fed DHA enriched diet. Based on this information we see potential in the further experiments on the cardiovascular system of Elovl2-/- mice.

Reviewer 3 Report

OVERALL ASSESSMENT. Although the topic of the present manuscript could be interesting, the experimental design and the statistical analysis are completely inaccurate. Therefore, I suggest a deep revision of the current form.

1.    No supplementary figures or tables are visible in the supplementary files, so is impossible understand some concepts.

2.    Why composition of LS group isn’t listed in Table 1? Please insert it. Moreover, explain in the methods, why FD&C dyes have been inserted in the diet and explain the acronym.

3.    In my opinion, the complete experimental procedures (administration of diets) are explained in confusing way in the methods section. Please insert a scheme for explain experimental procedures and move the explanation at lines 137-143 in the methods section. Moreover, what is the number of mice for each group? It must be inserted in the material and methods section together with the total number. The information at lines 55-56 is unclear and the declaration at line 87 as in the captions of all figures is useless. However, it is not scientific valid to compare experimental groups consisting of different number (3-8 as declared at line 87), so if one group is made of 3 mice, all the others must be randomly standardized to that number. In any case, for in vivostudy n=3 for each experimental group is a very low number. Finally, the same mice number must be maintained for all the analyses (line 202; why here only 3-4 instead of 3-8?). 

4.    Please move Fig.1 in the result section. Moreover, the use of the same symbols for different significant differences is really muddler. Please replace symbols with letters (for all the figures). Moreover, why the fifth group HS/HS isn’t included in this figure? Please insert also these data. Finally, some mistakes are present in this figure, since Wt-KO/LS and Wt-KO/LSDHA are never present in the legend, please correct.

5.    Because the authors compared more than two groups, t-student isn’t the correct statistical analysis. Please do the correct statistical analysis (e.g. ANOVA).

6.    Please move line 134 in the correct section.

7.    Table 2 isn’t well organized. It is divided in two different boxes. If the aim of this was to separate initial parameters from final parameters the final body weight must be moved in the lower box, and a third box must be created for the last three parameters. Finally, these results are useless without a statistical analysis that evaluates the statistically significant differences.

Author Response

Response to Reviewer 3 Comments

OVERALL ASSESSMENT. Although the topic of the present manuscript could be interesting, the experimental design and the statistical analysis are completely inaccurate. Therefore, I suggest a deep revision of the current form.

1.    No supplementary figures or tables are visible in the supplementary files, so is impossible understand some concepts. 

We apologize with the Reviewer for this oversight. In the revised manuscript, the data are presented clearly in the Supplementary file. 

2.    Why composition of LS group isn’t listed in Table 1? Please insert it. Moreover, explain in the methods, why FD&C dyes have been inserted in the diet and explain the acronym. 

We have modified Table 1 according to the Reviewer request and we have explained FD&C in the caption

3.    In my opinion, the complete experimental procedures (administration of diets) are explained in confusing way in the methods section. Please insert a scheme for explain experimental procedures and move the explanation at lines 137-143 in the methods section. Moreover, what is the number of mice for each group? It must be inserted in the material and methods section together with the total number. The information at lines 55-56 is unclear and the declaration at line 87 as in the captions of all figures is useless. However, it is not scientific valid to compare experimental groups consisting of different number (3-8 as declared at line 87), so if one group is made of 3 mice, all the others must be randomly standardized to that number. In any case, for in vivo study n=3 for each experimental group is a very low number. Finally, the same mice number must be maintained for all the analyses (line 202; why here only 3-4 instead of 3-8?). 

We have changed the method section according to the Reviewer suggestions. The new Figure 1 show the experimental set up with the complete number of mice we used.  

As the reviewer suggested we have changed the statistical analysis for Fig.2, Fig3 Fig5 and Fig.6. 

We used a random number generator in Excel to choose the n=4 (which was the smallest group) mice from the larger groups, so that all groups have the same number. After, we analyzed again the data using 2 ways ANOVA to see if there are any differences overall in the groups. If the p-value for the ANOVA was significant then we moved into individual t-tests comparing all of the 4 groups against one another (with Bonferroni correction).

We have changed in the method the “Statistical analysis” and in all the captions we wrote the same mice number we used.

4.    Please move Fig.1 in the result section. Moreover, the use of the same symbols for different significant differences is really muddler. Please replace symbols with letters (for all the figures). Moreover, why the fifth group HS/HS isn’t included in this figure? Please insert also these data. Finally, some mistakes are present in this figure, since Wt-KO/LS and Wt-KO/LSDHA are never present in the legend, please correct. 

- HS/HS control groups data are shown in Table 2. and in Fig.6 (gene expression).

- Figures legends have been corrected 

- We have changed the legends of the figures in order to make them more understandable.  

5.    Because the authors compared more than two groups, t-student isn’t the correct statistical analysis. Please do the correct statistical analysis (e.g. ANOVA).

We have corrected the statistical analysis in the manuscript.

6.    Please move line 134 in the correct section. 

We have done it

7.    Table 2 isn’t well organized. It is divided in two different boxes. If the aim of this was to separate initial parameters from final parameters the final body weight must be moved in the lower box, and a third box must be created for the last three parameters. Finally, these results are useless without a statistical analysis that evaluates the statistically significant differences. 

We have changed Table 2 according to the reviewer request and we added the statistical comments in the caption.

Round 2

Reviewer 1 Report

The authors response and the revised draft are fine. No other revision is needed on my side.

Author Response

Thank you for all your comments and questions. We are happy to hear that the manuscript is now suitable for the publication

Reviewer 3 Report

In present form the paper is acceptable. 

Author Response

Dear reviewer, 

thank you for all your comments and suggestions. According to the last comment of the Editor we have decided to keep the original number of mice. Results and statistics from the full number of animals originally treated in each group have been re-included in the paper. 
